# U-REPA: Aligning Diffusion U-Nets to ViTs

**Yuchuan Tian[1], Hanting Chen[2], Mengyu Zheng[3], Yuchen Liang[4], Chao Xu[1], Yunhe Wang[2]**

[1] State Key Lab of General AI, School of Intelligence Science and Technology, Peking University.
[2] Huawei Noah's Ark Lab. [3] The University of Sydney. [4] School of Mathematical Sciences, Peking University.
`tianyc@stu.pku.edu.cn`, {`chenhanting, yunhe.wang`}`@huawei.com`
`xuchao@cis.pku.edu.cn`

## Abstract

Representation Alignment (REPA) that aligns Diffusion Transformer (DiT) hidden-states with ViT visual encoders has proven highly effective in DiT training, demonstrating superior convergence properties, but it has not been validated on the canonical diffusion U-Net architecture that shows faster convergence compared to DiTs. However, adapting REPA to U-Net architectures presents unique challenges: (1) different block functionalities necessitate revised alignment strategies; (2) spatial-dimension inconsistencies emerge from U-Net's spatial downsampling operations; (3) space gaps between U-Net and ViT hinder the effectiveness of tokenwise alignment. To encounter these challenges, we propose **U-REPA**, a representation alignment paradigm that bridges U-Net hidden states and ViT features as follows: Firstly, we propose via observation that due to skip connection, the middle stage of U-Net is the best alignment option. Secondly, we propose upsampling of U-Net features after passing them through MLPs. Thirdly, we observe difficulty when performing tokenwise similarity alignment, and further introduces a manifold loss that regularizes the relative similarity between samples. Experiments indicate that the resulting U-REPA could achieve excellent generation quality and greatly accelerates the convergence speed. With CFG guidance interval, U-REPA could reach $FID < 1.5$ in 200 epochs or 1M iterations on ImageNet $256 \times 256$, and needs only **half** the total epochs to perform better than REPA under *sd-vae-ft-ema*. Codes: `https://github.com/YuchuanTian/U-REPA`

## 1 Introduction

Representation Alignment (REPA) [45], a methodology that aligns features from Diffusion Transformers (DiT) [30] to modern visual encoders, has been demonstrated to significantly accelerate DiT training. This approach holds particular significance given the growing prominence of DiTs, which have gained mainstream adoption in diffusion models and are extensively applied across image generation [4; 11; 23] and video generation domains [48; 22; 19]. However, emerging empirical evidence suggests that U-Net [33] architectures might present a more advantageous alternative to DiTs in certain scenarios [18; 7; 39; 38]: U-Net-based models exhibit substantially faster convergence while achieving generation quality comparable to their transformer-based counterparts. This dichotomy motivates our core research inquiry - can modern Vision Transformer (ViT [10])-based visual encoders be effectively adapted to guide diffusion U-Net training through alignment mechanisms similar to REPA, thereby potentially elevating the convergence speed ceiling of diffusion models?

However, establishing effective alignment between U-Net architectures and ViT-based encoders presents challenges. Unlike Diffusion Transformers (DiTs) that share structural similarities with Vision Transformers, U-Net architectures exhibit fundamentally different operational characteristics. Specifically, both DiT and ViT adopt isotropic architectures composed of uniformly stacked transformer blocks, which inherently facilitates straightforward parameter alignment between the two

39th Conference on Neural Information Processing Systems (NeurIPS 2025).

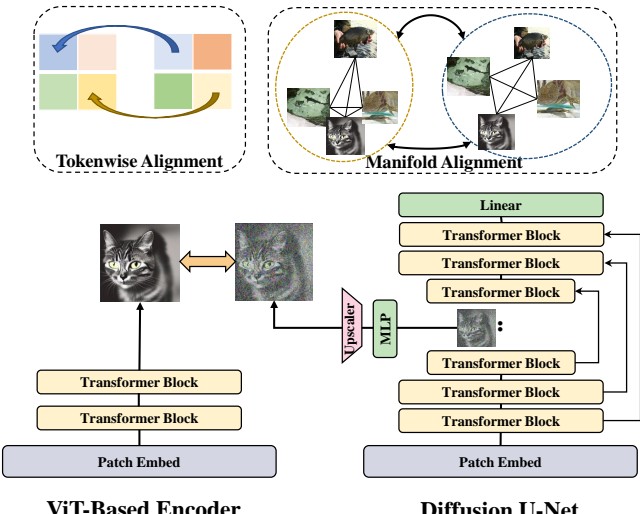

Figure 1: **The proposed U-REPA framework.** We investigated and found that semantic-rich intermediate layers are the best for representation alignment, dimension and space gaps hinders alignment efficacy. To counter these challenges, we scale-up features and propose manifold alignment.

frameworks. In contrast, U-Net's skip connections create strong interdependencies between shallow and deep network layers by linking them together, resulting in different feature propagation dynamics. This architectural disparity renders conventional representation alignment strategies developed for DiT architectures inapplicable to U-Net frameworks. Furthermore, the progressive downsampling operations in U-Net generate feature maps with spatial dimension mismatches compared to the fixed-scale feature representations in ViT encoders, introducing additional complexity in establishing cross-architectural correspondence. In addition, features from high-stage U-Net and ViT have large space gaps, forming a barrier for cosine similarities as metrics. Forcibly using tokenwise similarity as loss is not necessarily the best option. This induces us to rethink about the optimization objective.

In order to conquer these challenges, we propose **U-REPA**, a framework that aligns U-Net hidden states to features from ViT encoders. Firstly, Our analysis reveals that skip connections fundamentally alter the functional specialization of transformer blocks in U-Net architectures. By establishing direct dependencies between early-stage and late-stage layers, these cross-connections induce a hierarchical redistribution of semantic information, with intermediate blocks exhibiting the highest semantic density. This pattern was empirically verified through controlled ablation studies on DiT augmented with skip connections, where progressive layer-wise evaluations demonstrated peak semantic richness at median network depths.

The intermediate higher-stage layers, which contain semantically dense representations, require precise alignment with the ViT-based visual encoder. However, these critical layers undergo spatial downsampling in the U-Net architecture, necessitating explicit spatial dimension reconciliation between U-Net features and ViT features during representation alignment. Through empirical exploration of various resolution-matching strategies, we identified an optimal solution: performing linear transformation via MLP on U-Net features prior to upsampling operations, which achieves superior alignment performance compared to alternative approaches.

Further analysis revealed a fundamental incompatibility of measuring cosine similarity between the feature spaces of U-Net and ViT encoders. Enforcing strict token-wise similarity constraints proves excessively rigid due to inherent architectural discrepancies. To address this, we introduce a manifold loss that implements soft alignment through relational regularization. This loss operates on the relative geometric relationships between samples rather than imposing direct feature correspondence, thereby accommodating cross-architectural variations. Comprehensive experiments demonstrate that our proposed U-REPA framework effectively bridges the U-Net-ViT alignment gap while preserving the distinct advantages of both architectures.

Our contributions are as follows:

1. We identify U-Net's representation alignment to be a challenge due to different block functionalities, spatial-dimension inconsistencies, and larger feature space gaps.

2. We evaluate the contribution of downsampling and skips in U-Net and demonstrate U-Net's potential advantage over DiTs.

3. We propose U-REPA, a framework that evaluates layers, investigates the best scale-up policy, and introduces manifold-space loss in aid of alignment.

4. We conduct experiments and verify the effectiveness of our U-REPA framework in terms of fast convergence. Specifically, U-REPA reaches $FID < 1.5$ in just 200 epochs on ImageNet $256 \times 256$; and it reaches 1.41 FID, with only half the epochs of REPA under the same training setting.

## 2 Related Work

**The development of diffusion architectures.** The conventional diffusion works [17; 35; 36; 9] leverages a U-Net [33] architecture, whose basic block is a concatenation of convolution layers and self-attention. More recent architectural innovations in diffusion models have witnessed a paradigm shift from conventional U-Net frameworks toward transformer-based architectures. The emergence of Diffusion Transformers [1; 30] demonstrates their competitive performance despite abandoning the inductive biases inherent in U-Net designs. U-ViT [1] represents an intermediate architecture that preserves U-Net's hierarchical structure but replaces convolutional blocks with transformer layers, notably omitting the traditional downsampling operations. Subsequent developments have further streamlined the architecture: DiT [30] adopts a pure transformer backbone with isotropic scaling, while SiT [26] integrates the transformer architecture into the RectifiedFlow framework. Some other works either improve the micro-designs [6; 25], or focuses on architectural efficiency [3; 42; 40].

In contrary to these DiT works, some works still sticks to U-Net architectures and offer valuable rethinking on this conventional architectural preference: in pixel-space image generation, works including SimpleDiffusion [18] and HourglassDiT [7] still sticks to U-Net; in with its variants like U-DiT [39], Playground v3 [24], and DiC [38] extending its success to latent-space diffusion through simple Conv3×3 designs. While these implementations empirically validate U-Net's accelerated convergence and stable training dynamics compared to transformer-based alternatives, current research predominantly focuses on proposing architectural modifications rather than uncovering the reasons of U-Net's superior diffusion performance.

**Techniques for better DiT performance.** Building upon the success of self-supervised learning [16], MDT [13] and MaskDiT [47] pioneer masked image modeling in diffusion frameworks by adaptively masking a good proportion of input patches during training. Other than the masking strategy, a bunch of diffusion works refer to higher-level semantic guidance from off-the-shelf pretrained models that significantly improves generation quality. REPA [45] establishes feature alignment between ViT-based encoder embeddings and diffusion latent spaces through contrastive learning. LightningDiT [44] innovates through an improved VAE distilled from MAE [16] and DINOv2 [28]. Ma et al. [27] introduces CLIP [31] and DINO [2] to verify inference-time scaling of diffusion models. These methods demonstrate that a higher-level semantic-rich feature-map from pretrained vision encoders is helpful to diffusion-based generation.

## 3 Method

### 3.1 Preliminaries: REPA for DiT

Representation Alignment (REPA) [45] distills Diffusion Transformers with semantic features from off-the-shelf ViT-based vision encoders (e.g. DINOv2 [28], CLIP [31], MAE [16], et cetera). Given a ViT-based vision encoder $f$ and clean image $\mathbf{x}_*$, let $\mathbf{y}_* = f(\mathbf{x}_*) \in \mathbb{R}^{N \times D}$ denote its patch embeddings, where $N$ and $D$ represent the number of patches and embedding dimension, respectively. REPA establishes feature alignment between $\mathbf{y}_*$ and the projected diffusion encoder outputs $h_\phi(\mathbf{h}_t)$, where $\mathbf{h}_t = f_\theta(\mathbf{z}_t)$ is the latent representation from the diffusion transformer at timestep $t$, and $h_\phi$ is a trainable multilayer perceptron (MLP).

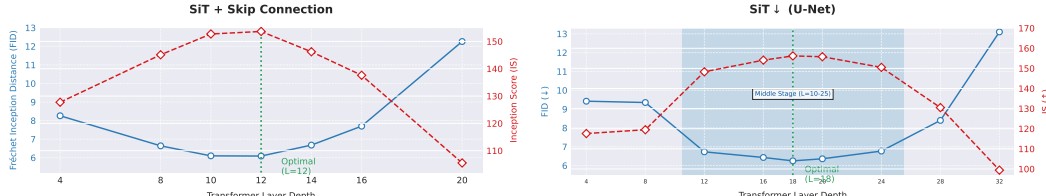

Figure 2: **Investigating alignment with respect to encoder depths on diffusion models with skip connections. Left:** SiT with skip connections. Due to the change of block functionalities due to newly established skip dependencies, the most optimal encoder depth is shifted towards the middle of the model. **Right:** SiT↓, the U-Net-based SiT model. Shadowed region represents higher U-Net stage. The plot infers that stage transitions (downsampling& upsampling in U-Net) bring large block functionality gaps. Alignment within higher U-Net stage is thus necessary for alignment performance.

The alignment is enforced by maximizing token-wise feature similarities, *i.e.* the similarity of a token from DiT hidden-state with its corresponding counterpart in the ViT encoder feature:

$$\mathcal{L}_{\text{REPA}}(\theta, \phi) := -\mathbb{E}_{\mathbf{x}_*,\epsilon,t}\left[\frac{1}{N}\sum_{n=1}^{N}\text{sim}\left(\mathbf{y}_*^{[n]}, h_\phi(\mathbf{h}_t^{[n]})\right)\right], \tag{1}$$

where $\text{sim}(\cdot, \cdot)$ denotes a similarity metric (e.g., cosine similarity). Typically, $\mathbf{z}_t$ is adopted as the output from early layers (the original work adopts layer index 8) in DiT for better alignment. This alignment term is combined with the basic flow-based diffusion objective (*i.e.* SiT [26]) through a tunable coefficient $\lambda > 0$, and the final loss for diffusion model training is formulated as follows:

$$\mathcal{L} := \mathcal{L}_{\text{velocity}} + \lambda\mathcal{L}_{\text{REPA}}. \tag{2}$$

## 3.2 Evaluating the Potential of U-Net

In diffusion models, U-Net and isotropic architectures (e.g., DiT) exhibit distinct design philosophies. While DiT achieves state-of-the-art results through scalability and integration with advanced techniques, U-Net-based methods emphasize faster convergence [39]. To dissect U-Net's efficacy, we isolate its two core components: skip connections and downsampling.

1. **Skip Connections:** Provide shortcuts between encoder and decoder layers, theoretically aiding gradient flow and feature reuse.
2. **Downsampling:** Reduces spatial resolution (typically by a scale factor of 2 at each stage) to enable hierarchical, multi-scale feature learning. Critically, downsampling is always paired with skip connections to mitigate information loss.

**Toy experiments on U-Net components.** On top of DiT, we perform toy experiments that reveal the contribution of components mentioned above.

| ImageNet 256×256, DiT 400K, cfg=1 | | | |
|---|---|---|---|
| Model | FLOPs (G) | FID↓ | IS↑ |
| **DiT-XL/2** | 118.6 | 19.47 | - |
| **DiT-XL/2***  | 118.6 | 20.05 | 66.74 |
| **+ Skip Connections** | 114.1 | 19.86 | 67.29 |
| **+ Downsampling** | 108.8 | 13.78 | 88.93 |
| **DiT↓-XL/2 (+Tricks)** | 108.8 | **11.02** | **100.35** |

Table 1: **Evaluating the contribution of U-Net components in terms of fast convergence.** Experiments are conducted using hyperparameters from [30] for 400K iterations. Model depth is changed when a modification is made such that the overall FLOPs is kept almost the same with DiT.

This suggests that U-Net's fast-convergence advantages primarily stem from multi-scale hierarchical modeling via downsampling, not skip connections. Downsampling compresses features into compact, semantically rich representations, accelerating learning while maintaining information flow through skip-augmented decoder layers. However, skip connection is not useless as it compensates for the information loss due to downsampling.

Building on this insight, we propose DiT↓ (and SiT↓ for the flow-based version, following the naming convention of [26]) by adding tricks of RoPE [37] and SwiGLU following previous work [6; 39]

### 3.3 Aligning U-Net to ViT Encoders

Since U-Net has good potentials to achieve excellent generation, we are motivated to investigate whether REPA could also work on U-Net. We are first focused on the block functionality pattern and investigates the most optimal position for alignment; then we are interested in feature size alignment problems; lastly, we are dedicated to merging space gaps between features from U-Net and ViT, respectively.

**Position for alignment.** Regarding the concern that block functionalities differ between U-Net and DiT, the comparison of prior studies [12; 5] demonstrates divergent hierarchical specialization: U-Net architectures typically employ mid-network layers for high-level semantic synthesis while reserving shallow layers for low-level image refinement, whereas DiTs exhibit a totally different pattern - early layers primarily govern semantic-rich outline formation with deeper layers handling detailed image refinement.

These previous findings find empirical support in REPA's experimental findings, where representation alignment proves most effective when applied to initial transformer blocks. This phenomenon stems from DiT's early layers encoding semantic-rich representations that align well with the semantically dense outputs of ViT-based visual encoders, enabling meaningful guidance. Unlike the straightforward DiT architecture, the inherent skip connections in U-Net architectures induce fundamentally distinct block functionality compared to Diffusion Transformers. While all blocks in ViT or DiT maintain homogeneous computational roles, following a continuous flow of transition from input to output, U-Net's cross-layer shortcuts establish direct dependencies between shallow and deep layers, fundamentally altering feature-map evolution patterns. As shown in Fig. 2 (R), DiTs with skips indicates median layer is the best for representation alignment. The same pattern goes for U-Net (Fig. 2 (L)) despite the downsampling stage.

**Feature size alignment.** The implementation of alignment between Diffusion U-Net's median stage and ViT encounters a critical spatial resolution dilemma stemming from architectural disparities. While our analysis identifies mid-network U-Net features as optimal semantic carriers, their spatial dimensions drastically differ from ViT's full-resolution token sequence. This dimensional mismatch obstructs REPA's token-wise similarity computation, which requires strict cardinality matching between compared features.

In order to align two feature-maps (*i.e.* from U-Net and ViT encoder, respectively), from the macro level we advocate for upscaling the smaller-sized U-Net features rather than downscaling the larger visual encoder features. This design principle stems from the critical observation that compressing ViT's high-resolution features to match U-Net's reduced dimensions inevitably discards fine-grained visual information, thereby degrading alignment effectiveness. Preserving ViT's native resolution while expanding U-Net's bottleneck features proves essential for maintaining semantic fidelity.

At the implementation level, we empirically evaluated various upscaling strategies for U-Net features:

1. **Upscale first and then MLP:** feature upsampling is performed before passing the feature into the MLP.

2. **Upscale within MLP:** the MLP also acts as a feature upsampler that receives a low-resolution input and outputs a high-resolution one via linear mapping and pixel un-shuffling.

3. **MLP first and then upscale:** the feature from higher-stage U-Net is first passed through MLP and then upsampled.

Among the three options, we found "MLP first and then upscale" is the best both in terms of performance and efficiency (minimum FLOPs cost), which will be discussed in the Ablation Study in Sec. 4.

**Manifold space alignment.** Though we select the most suitable U-Net feature for alignment and keep dimensions between U-Net and ViT features aligned, challenges remain in feature space compatibility. First, compared to the structural congruence between DiT and ViT encoders, the architectural discrepancy of U-Net (with its skip connections and hierarchical downsampling) creates a more pronounced feature distribution gap between U-Net hidden states and visual encoder outputs.

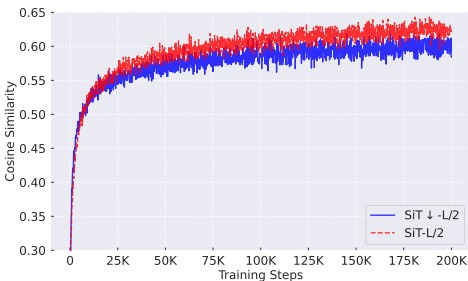

Figure 3: **The convergence of average tokenwise similarities.** While SiT-L/2 could achieve better tokenwise similarities, SiT↓ converges at a lower similarity value, indicating difficulties of feature alignment.

Second, the dimensional transformation required for alignment inevitably modifies U-Net's native feature space characteristics.

Some recent works on Diffusion U-Net [41; 34] reveals that higher-stage U-Net features are low-frequency subspaces that discards higher frequency compoents, including noises. Gaps are inevitable when evaluating the cosine similarities of detail-rich, high-frequency-rich vectors and flat, low-frequency-dominated vectors. In this sense, strict token-level alignment constraints like the original REPA loss prove suboptimal under these conditions, as they assume implicit feature space homogeneity between aligned modalities.

As is shown in Fig. 3, we conducted continuous measurements of token-wise cosine similarity against ViT features during training. The two models that we compare are SiT-L/2 of isotropic, standard transformer architecture and SiT↓-L/2 of U-Net architecture. Our experiments revealed a characteristic learning trajectory: while U-Net achieves slightly faster similarity improvement in early training phases - thanks to skip connection that helps convergence - its progress stagnates beyond this point, ultimately plateauing at 0.60 - notably inferior to DiT's sustained growth reaching around 0.63 similarity. The similarity gap between SiT and SiT↓ This phenomenon suggests that naively aligning U-Net with ViT encoders through angular similarity metrics alone encounters inherent limitations due to architectural incompatibilities.

Rather than strict token-wise regularization, we resort to looser objectives that does not require rigid augular alignment. Inspired by manifold knowledge distillation [14], we hold that aligning similarities between samples from the same feature space could be a promising solution. Hence, we define Manifold Loss $\mathcal{L}_{\text{ML}}$ as

$$\mathcal{L}_{\text{ML}}(\theta, \phi) := -\mathbb{E}_{\mathbf{x}_*, \epsilon, t, i, j} \left[ \text{d}(\mathbf{y}_*, h_\phi(\mathbf{h}_t)) \right], \tag{3}$$

where

$$\text{d} := \|\text{sim}\left(\mathbf{y}_*^{[i]}, \mathbf{y}_*^{[j]}\right) - \text{sim}\left(h_\phi(\mathbf{h}_t^{[i]}), h_\phi(\mathbf{h}_t^{[j]})\right)\|_F^2. \tag{4}$$

In the formula, cosine similarity is adopted as the similarity metric, and $F$ represents Frobenius Norm of matrices. By introducing affine hyperparameter $w$, the overall optimization target is then formulated as

$$\mathcal{L} := \mathcal{L}_{\text{velocity}} + \lambda \left( \mathcal{L}_{\text{REPA}} + w \mathcal{L}_{\text{ML}} \right). \tag{5}$$

## 3.4 Other Improvements

We also propose and evaluate some other improvements. Due to page limits, the proposed methods and corresponding ablations are enclosed in the Appendix.

# 4 Experiments

## 4.1 Experiment Setup

**Experiment settings.** Our implementation completely adheres to the training protocol established in REPA [45]. Following the architectural configuration of latent diffusion models [32], we employ the identical VAE variant (*sd-vae-ft-ema*) and adopt the AdamW optimizer. To ensure fair comparison, we maintain identical hyperparameter settings across all experiments: a global batch size of 256, fixed learning rate of $1e-4$, and disabled weight decay (set to 0). $(\beta_1, \beta_2)$ is set as (0.9, 0.999). All experiments are conducted on the ImageNet 2012 benchmark [8] under a controlled environment with a fixed random seed (global seed=0). 8 NVIDIA A100 GPUs are used for main experiments.

For main experiments (Tab. 5), we apply guidance interval [21] $[0, 0.7]$ and SDE sampling according to the convention of REPA [45] for fair comparison. We select smaller cfg of 1.65, because we found it is better for our architecture, different from SiT. For all ablation experiments, we train models for 100K iterations, which is sufficient to show the trend of model performance; sampling is conducted with the default setting of the official REPA codebase, *i.e.* $cfg = 1.8$ in ODE and guidance interval $[0, 0.7]$.

**Model settings.** By aligning channel dimensions and FLOPs with standard Diffusion Transformers (DiTs or SiTs), our U-REPA-compatible variants maintain architectural parity while introducing critical adaptations for U-Net principles. The base model (SiT↓-B) employs a stage arrangement of [5,5,5], achieving 199.7G FLOPs. Scaling to larger models, we have the L variant (686.6M) and XL variant (954.4M params) that increases channel width (1024 vs. 1152 in base) through increased stage-wise block allocation ([9,14,9] vs. [10,16,10]). Notably, when FLOPs are aligned, SiT↓ models usually have more parameters than SiTs due to increased depth.

| Model | Params (M) | FLOPs (G) | Patch Size | Channel | # Heads | Blocks in Stages |
|---|---|---|---|---|---|---|
| **SiT↓-B** | 199.7 | 24.1 | 2 | 768 | 12 | [5,5,5] |
| **SiT↓-L** | 686.6 | 79.3 | 2 | 1024 | 16 | [9,14,9] |
| **SiT↓-XL** | 954.4 | 109.3 | 2 | 1152 | 16 | [10,16,10] |

Table 2: **Configurations of SiT↓ architecture at different model sizes.** The proposed SiT↓ in U-Net architectures are aligned to DiTs in terms of FLOPs and channel dimension.

## 4.2 The Advantage of U-REPA

**Comparing SiT↓ with SiT at different scales.** We evaluate our U-REPA alignment method on ImageNet 256 under a generation setting with $cfg = 1$ (REPA framework without classifier-free guidance). As shown in Table 3, our approach consistently improves generation quality while significantly reducing computational costs across model scales. For the base-size SiT-B/2 variant, integrating U-REPA achieves a 39.3% improvement in FID (from 24.4 to 15.3) with comparable FLOPs (24.1G vs. 23.0G) and identical training iterations (400K), demonstrating that feature alignment enhances parameter efficiency without additional training overhead. The acceleration effect becomes more pronounced in larger models: for SiT-L/2, U-REPA reduces required iterations by 42.9% (700K→400K) while simultaneously lowering FLOPs (79.3G vs. 80.8G) and achieving a 30.9% FID improvement (8.4→5.8). Most notably, the XL-scale variant with U-REPA (*cf.* Fig. 5 for FIDs vs. Training iters) attains state-of-the-art FID (5.4) using **90%** fewer iterations (400K vs. 4M) and fewer FLOPs (108.8G vs. 118.6G) compared to the baseline, proving our method's fast convergence.

We also demonstrate the advantage of the proposed U-REPA framework when measuring by parameters (rather than computation FLOPs), as shown in Fig. 4. Though U-Net brings extra parameters when FLOPs are aligned with DiTs, the advantage of SiT↓+U-REPA is obvious as depicted in the Parameter versus FID plot.

**Convergence performance.** We also compare our method with previous State-of-the-Arts, as shown in Tab. 5. Our proposed SiT↓+U-REPA achieves a competitive FID of 1.48 with only 200 training epochs, significantly outperforming existing methods in training efficiency. Notably, while state-of-the-art masked diffusion transformers like MDTv2-XL/2 require 1,080 epochs to reach 1.58 FID, our method attains better performance (1.48) with 80% fewer iterations. Even compared to the SOTA

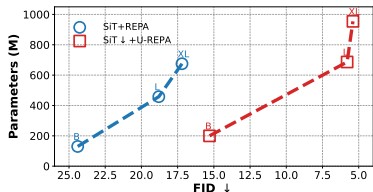

Figure 4: **Comparing SiT↓+U-REPA against SiT+REPA in terms of parameter scalability.** While the U-Net architecture makes the diffusion model parameter-rich compared with same-FLOPs Diffusion Transformers, SiT↓ models still outcompetes SiTs by large margins in terms of parameters.

| ImageNet 256×256, w/o cfg | | | |
|---|---|---|---|
| Model | FLOPs (G) | Iter. | FID↓ |
| **SiT-B/2+REPA** | 23.0 | 400K | 24.4 |
| **SiT↓-B/2+U-REPA** | 24.1 | 400K | **15.3** |
| **SiT-L/2+REPA** | 80.8 | 700K | 8.4 |
| **SiT↓-L/2+U-REPA** | 79.3 | 400K | **5.8** |
| **SiT-XL/2+REPA** | 118.6 | 4M | 5.9 |
| **SiT↓-XL/2+U-REPA** | 108.8 | 400K | **5.4** |

Table 3: **Comparing U-REPA against REPA across various model sizes without classifier-free guidance.** U-Nets equipped with U-REPA show excellent capabilities. Notably, U-REPA achieves 10× faster convergence compared with REPA in terms of performance w/o CFG.

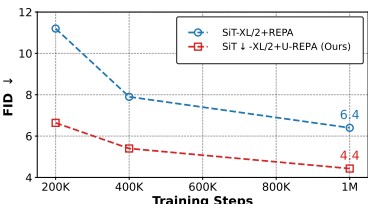

Figure 5: **Comparing SiT↓+U-REPA against SiT+REPA in terms of convergence speed.** SiT↓-XL/2 convergences much faster than SiT-XL/2 with the help of U-REPA.

| ImageNet 256×256, w/ cfg | | | | |
|---|---|---|---|---|
| Model | Dep. | Feat. Dim. | FID↓ | IS↑ |
| **SiT↓-XL/2** | 4 | 16 | 9.42 | 117.6 |
| **SiT↓-XL/2** (REPA) | 8 | 16 | 9.35 | 119.5 |
| **SiT↓-XL/2** | 12 | 8 | 6.73 | 148.3 |
| **SiT↓-XL/2** | 16 | 8 | 6.43 | 154.1 |
| **SiT↓-XL/2** | **18** | **8** | **6.25** | **156.2** |
| **SiT↓-XL/2** | 20 | 8 | 6.36 | 155.8 |
| **SiT↓-XL/2** | 24 | 8 | 6.77 | 150.5 |
| **SiT↓-XL/2** | 28 | 16 | 8.40 | 130.5 |
| **SiT↓-XL/2** | 32 | 16 | 13.10 | 99.4 |

Table 4: **Ablations on encoder depths for alignment in SiT↓.** Feat. Dim. stands for the spatial height & width at the certain layer. Compared with the default REPA setting, aligning at the centering layer (higher stage in U-Net) performs much better.

SiT-XL/2 + REPA baseline (800 epochs for 1.42 FID), our approach uses only 1/2 of the training epochs (400) while achieving better generation quality (1.41 FID). The results demonstrate that the proposed U-REPA establishs a new efficiency frontier for diffusion models.

## 4.3 Ablation Studies

**Encoder depths.** The ablation study on encoder layer depths for feature alignment (Tab. 4) coincides with the pattern of DiT with skip connections, as we analyzed in Sec. 4: despite progressive down-sampling operations that reduce spatial resolution, the centermost layers exhibit optimal alignment efficacy. For the SiT↓-XL/2 model, aligning features at layer 18 (midway through the 36-layer architecture) achieves peak performance with 6.25 FID and 156.2 IS, outperforming both shallower and deeper alignment points. This phenomenon persists even as the spatial dimension (Feat. Dim.) halves from 16×16 to 8×8 in the intermediate stage, indicating that semantic richness—not spatial resolution—dominates alignment quality. Performance degradation occurs when alignment takes place at shallower or deeper stages, even though the feature size is kept the same with DINO in these stages.

**Alignment dimension choices.** The comparative results in Table 6 reveal that upsampling U-Net's higher-stage features ($\uparrow_2$) to match DINOv2's native resolution achieves superior performance (5.72 FID, 161.6 IS), outperforming the alignment alternative in generation quality. This demonstrates that preserving ViT encoder's original feature granularity during alignment is beneficial for alignment.

**Feature-map upscale choices.** Among the three upscaling options mentioned in Sec. 4, we figure out that upscaling U-Net hidden states after getting passed through MLP is the best option, achieving 5.72 FID and 161.6 IS. This option is also the most optimal one in terms of computation cost analysis.

**ImageNet 256×256, w/ cfg**

| Alignment Choices | FID↓ | IS↑ |
|---|---|---|
| **U-Net ‖ DINOv2$_{\downarrow 2}$** | 5.99 | 158.8 |
| **U-Net$_{\uparrow 2}$ ‖ DINOv2** | **5.72** | **161.6** |

Table 6: **Alignment dimension choices.** Up-sampling higher-stage U-Net features in alignment with ViT performs better due to less information loss.

**ImageNet 256×256, w/ cfg**

| Alignment | FID↓ | IS↑ |
|---|---|---|
| **Upscale before MLP** | 5.84 | 158.5 |
| **Upscale in MLP** | 6.36 | 153.4 |
| **Upscale after MLP** | **5.72** | **161.6** |

Table 7: **Feature-map upscale choices.** Among the three options, Upscaling after passing through MLP performs best; and it has lower cost as the small-sized feature map is passed through MLP.

**ImageNet 256×256, w/ cfg**

| Model | $w$ | FID↓ | IS↑ |
|---|---|---|---|
| **SiT↓-XL/2+U-REPA** | 0 | 6.25 | 156.2 |
| **SiT↓-XL/2+U-REPA** | 2 | 5.81 | 160.8 |
| **SiT↓-XL/2+U-REPA** | **3** | **5.72** | **161.6** |
| **SiT↓-XL/2+U-REPA** | 4 | 5.79 | 160.6 |

Table 8: **Adjusting weight $w$ in Eq. 5.** Manifold loss boosts U-Net's alignment performance. The most optimal result is taken at $w = 3$.

**ImageNet 256×256, w/ cfg**

| Model | Epochs | FID↓ |
|---|---|---|
| *Pixel diffusion* | | |
| ADM-U [9] | 400 | 3.94 |
| VDM++ [20] | 560 | 2.40 |
| Simple diffusion [18] | 800 | 2.77 |
| *Latent Diffusion Transformer* | | |
| U-ViT-H/2 [1] | 240 | 2.29 |
| DiffiT [15] | - | 1.73 |
| DiT-XL/2 [30] | 1400 | 2.27 |
| SiT-XL/2 [26] | 1400 | 2.06 |
| *Masked Diffusion Transformer* | | |
| MaskDiT [47] | 1600 | 2.28 |
| MDTv2-XL/2 [13] | 1080 | 1.58 |
| *Representation Alignment* | | |
| SiT-XL/2 + REPA [45] | 800 | 1.42 |
| SiT↓-XL/2 + U-REPA (Ours) | 200 | 1.48 |
| **SiT↓-XL/2 + U-REPA (Ours)** | **400** | **1.41** |

Table 5: **Comparing U-REPA against State-of-the-Art baselines with classifier-free guidance.** U-REPA could reach $FID < 1.5$ in merely 200 epochs and $FID = 1.41$ in 400 epochs; The proposed method converge $2\times$ faster while achieving lower FID.

**Manifold loss weight** $w$. The ablation study on alignment weight $w$ in Eq. 5 demonstrates a clear performance peak at $w = 3$ achieving the lowest FID (5.72) and highest IS (161.6) among tested configurations.

## 4.4 Higher Resolution Experiments

At the higher-resolution ImageNet 512×512 (w/ cfg) setting, U-REPA remains clearly superior to the REPA baseline (Tab. 9). Using SiT↓-XL/2, U-REPA reduces FID from 2.44 to 2.21 and raises IS from 247.3 to 274.7. These results indicate that U-REPA's benefits persist at 512 resolution, yielding better distributional fidelity and sample quality/diversity, and demonstrating strong scalability.

**ImageNet 512×512, w/ cfg**

| Alignment Choices | FID↓ | IS↑ |
|---|---|---|
| SiT-XL/2 + REPA | 2.44 | 247.3 |
| **SiT↓-XL/2 + U-REPA (Ours)** | **2.21** | **274.7** |

Table 9: **Comparing U-REPA against REPA on ImageNet 512×512.** On higher resolution, the proposed U-REPA still maintain a clear advantage.

## 4.5 The Energy Cost Advantage of U-REPA

We also assess the energy-cost advantage of U-REPA over REPA. We train on eight NVIDIA A100 GPUs and record each GPU's power draw. Combining the measured power with the training duration,

| ImageNet 256×256 | | | | |
|---|---|---|---|---|
| Model | Avg. Pow. (W) | Training Hours | Est. Energy (J) | FID↓ |
| **SiT-XL/2+REPA** (4M iter) | 373.2 | 302.7 | 3.25×10e9 | 1.42 |
| **SiT↓-XL/2+U-REPA** (2M iter) | **295.3** | **230.7** | **1.96×10e9** | **1.41** |

Table 10: **Energy cost comparison.** We compare the energy cost of U-REPA (at 2M iters) and REPA (at 4M iters). U-REPA could significantly reduce the cost of training a State-of-the-Art diffusion model.

we estimate the total energy consumed. The statistics for average power and estimated total energy used by all 8 GPUs are summarized in Tab. 10. Results indicate that our U-REPA method is "greener", costing far less energy.

Reducing training energy directly curbs operational $CO_2$ emissions. Methods that achieve comparable accuracy with lower energy, such as U-REPA vs. REPA in our study, advance both sustainability and the economic viability of large-scale AI.

## 5   Conclusion

In this paper, we propose U-REPA, an adapted version of REPA on Diffusion U-Net. We identify key challenges in U-Net hidden state alignment and show that U-REPA effectively bridges the gap between U-Net-based diffusion models and ViT-based encoders. By aligning intermediate features, resolving spatial mismatches via post-MLP upsampling, and enforcing manifold-aware regularization, U-REPA achieves faster convergence and an FID score of 1.41 on ImageNet-256×256 at 2M iters.

**Acknowledgement.** This work is supported by the National Key R&D Program of China under grant No. 2022ZD0160300 and the National Natural Science Foundation of China under grant No. 62276007. This work is funded by Peking University–BHP Carbon and Climate Wei-Ming PhD Scholars Program (Program Name: Research on Low-Carbon and Energy-Efficient Large Model Architectures; Program Number: WM202505). We sincerely thank Sibo Fang for his generous help during this project.

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

## Appendix

## A  Other Improvements

In the appendix, we address the effect of proposed "Other Improvements" (Sec. 4), including "Time-aware MLP" and "Weight schedule of loss".

**Time-aware MLP.** Some works on in the diffusion task [29; 46] reveals that a channel dimension is particularly sensitive and useful to a certain subset of time during sampling. As the time-variant feature is to be aligned to time-invariant vision encoder features, we hold that alignment could perform better when the MLP is time-aware and extract time-invariant information out of diffusion features for alignment.

Inspired by the conditioning of canonical Diffusion U-Net [17; 36; 9] and DiT [30], we add a module to predict a pair of channel-wise shift& scale vector $(\gamma(t), \beta(t))$. The module is in parallel to MLP and follows the design of DiT's AdaLN, which is a concatenation of a SiLU and a Linear layer. The shift& scale vectors are imposed on the output MLP as follows:

$$h_\phi^t(\mathbf{h}_t^{[n]}) = \gamma(t) \odot h_\phi(\mathbf{h}_t^{[n]}) + \beta(t)$$

**Weight schedule of loss.** As is prompted in REPA [45], designing weight schedule is a future direction. We try various weight schedules (*i.e.* make $\lambda$ in Eq. 2 a function $\lambda(t)$ with respect to time) but found that these schedules bring very limited improvements on the proposed U-REPA. Hence, we stick to the original constant weight strategy of REPA.

These improvements bring slight increases on the generation metrics. Hence, we do not include them in the main experiments for the simplicity of the method. The effects of proposed measures are shown in Tab. 11 and Tab. 12.

| ImageNet 256×256, w/ cfg | | |
|---|---|---|
| Alignment Choices | FID↓ | IS↑ |
| **Ordinary MLP** | 5.72 | 161.6 |
| **Time-aware MLP** | **5.63** | **163.3** |

Table 11: **The effect of time-aware MLPs.**

| ImageNet 256×256, w/ cfg | | |
|---|---|---|
| Alignment Choices | FID↓ | IS↑ |
| **Constant** | 5.72 | 161.6 |
| $\mathbf{max}(1, t + 0.5)$ | 5.85 | 161.3 |
| $\mathbf{max}(1, -t + 1.5)$ | 5.72 | 161.6 |
| $\mathbf{min}(1, \mathbf{max}(-2t + 1.5, 2t - 0.5))$ | **5.58** | **164.0** |

Table 12: **The effect of different weight schedules of loss.**

## B  Additional Experiments

**Evaluating SiT↓-XL/2.** We also evaluated the proposed U-Net architecture on the Scalable Interpolant Transformers (SiT) framework without the guidance of REPA. The results are shown in Tab. 13.

Notably, though the amount of FID improvement brought by U-REPA is not as great as REPA (i.e. SiT / SiT+REPA vs. SiT↓+U-REPA), we hold that this comparison is invalid due to the following reasons:

1. As generation performance gets stronger, it is also becoming much harder to improve (especially for FID when it gets lower).

2. Aligning SiT↓ and ViT is much harder than aligning SiT and ViT, because the backbone of SiT and ViT encoders are very similar. Aligning SiT↓ to ViT encoder is a special case due to great architecture difference.

However, we hold that comparing REPA and U-REPA on the same model of SiT↓ is fair. The default REPA achieves FID 9.35 (as shown in Tab. 4) while our method achieves FID 5.72, both trained for 100K iterations with cfg and guidance interval adopted.

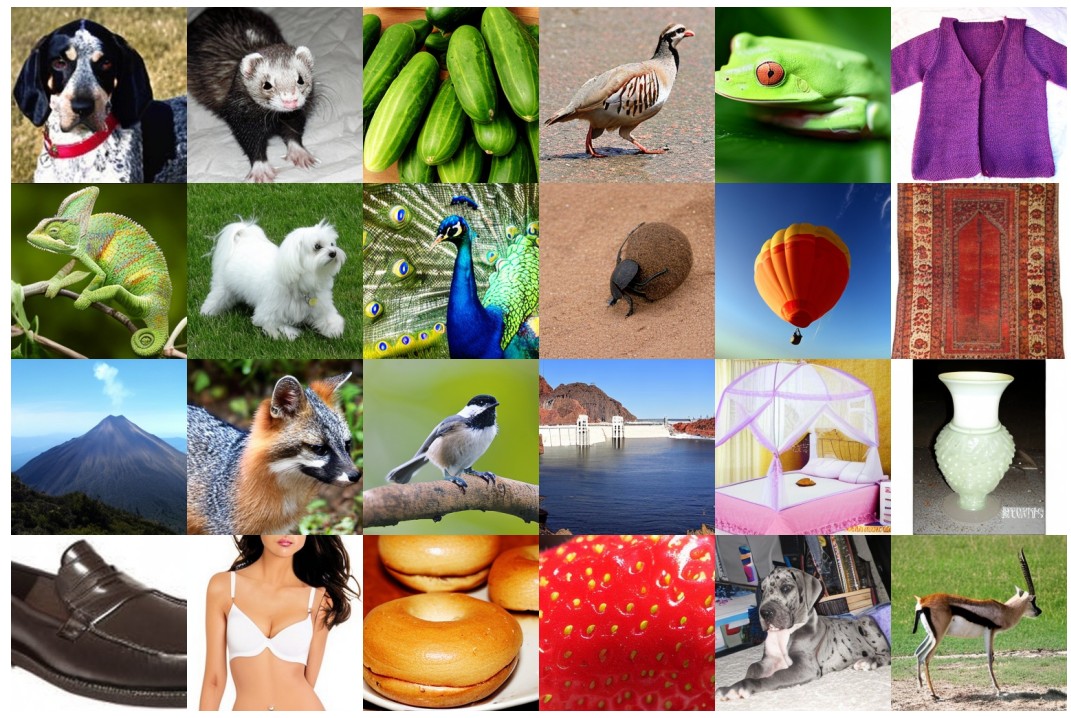

Figure 6: **Samples generated by SiT↓+U-REPA at 1M iterations.** The samples are generated following the setting of REPA, at $cfg = 4$. Best viewed on screen.

| ImageNet 256×256, w/o cfg | | |
|---|---|---|
| Model | Iter. | FID↓ |
| **SiT-XL/2** | 400K | 17.2 |
| **SiT↓-XL/2** | 400K | 9.2 |
| **SiT-XL/2+REPA** | 400K | 7.9 |
| **SiT↓-XL/2+U-REPA** | 400K | **5.4** |

Table 13: **Evaluating the performance of SiT↓.** SiT↓-XL/2 performs much better than SiT-XL/2, and U-REPA further reduces the FID of SiT↓-XL/2 to 5.4 without classifier-free guidance.

**Seed Sensitivity.** In our paper, we take $seed = 0$ following the setting of REPA. We also tested other seeds ($seed = 1, 2$) in training to examine the seed sensitivity of our method, shown in Tab. 14. The experiments are run for 600K iterations with guidance interval and cfg, following REPA.

| ImageNet 256×256, w/ cfg | | |
|---|---|---|
| Model | seed | FID↓ |
| **SiT↓-XL/2+U-REPA** | 0 | 1.618 |
| **SiT↓-XL/2+U-REPA** | 1 | 1.599 |
| **SiT↓-XL/2+U-REPA** | 2 | 1.588 |
| **SiT↓-XL/2+U-REPA** | mean | 1.602±0.012 |

Table 14: **Examining seed sensitivity.** We selected $seed = 0, 1, 2$ and evaluate the performance with cfg. The performance fluctuation is limited to a narrow interval (approximately 0.01).

**Ablations on the REPA Loss.** We also conduct ablations on the REPA loss ($\mathcal{L}_{REPA}$) while leaving the proportion of manifold loss intact (keeping the multiplication $\lambda w$ fixed). The results are shown in Tab. 15.

| ImageNet 256×256, w/ cfg | | | |
|---|---|---|---|
| $\lambda$ | **0.5** | 0.25 | 0 (No $\mathcal{L}_{REPA}$) |
| FID↓ | **5.72** | 6.42 | 10.91 |

Table 15: **Adjusting the hyperparameter for REPA loss $\lambda$ in Eq. 5.** The REPA loss is vital for the generation performance; removing $\mathcal{L}_{REPA}$ would cause a significant performance decay.

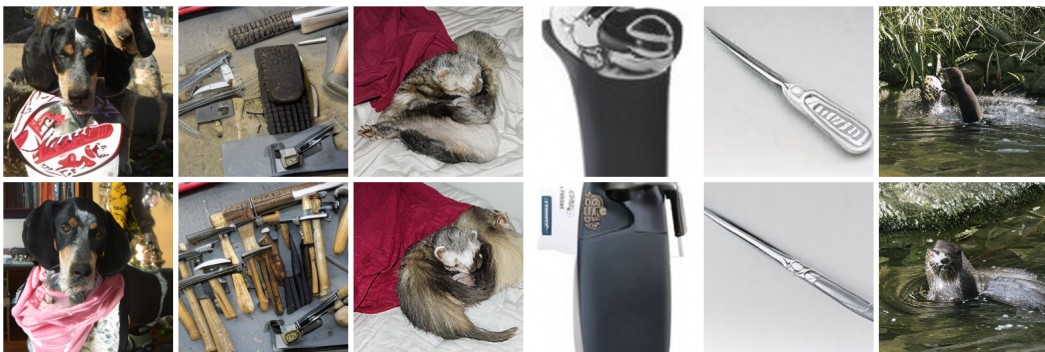

Figure 7: **Comparing the visual quality of SiT+REPA (upper row) and SiT↓+U-REPA (lower row).** The samples are generated following the sampling strategy that yields the State-of-the-Art FIDs in respective methods. Best viewed on screen.

**One-on-one visualization comparison.** Apart from quantitative comparisons, we also provide qualitative comparisons in Fig. 7 by inserting the same rnadom noise into trained SiT+REPA (at 4M iterations, FID 1.42) and SiT↓+U-REPA (at 2M iterations, FID 1.41). The samples are not cherrypicked; we directly pick the first several samples at seed=0. Samples generated by SiT↓+U-REPA has better visual quality.

## C   Limitations & Impact

**Limitations and Future work.** The U-Net architecture is a simple one with only one intermediate stage. We do not further refine the architecture as we want to show U-Net architectures as simple as SiT↓ could also achieve rapid convergence. Further improvements on the U-Net architecture includes efficient attention [42; 39], non-integer down& up scaling factors [43], and more use of convolutions [36; 38]. Besides, whether U-REPA could be applied to downstream diffusion tasks that rely heavily on U-Nets (e.g. Low-Level Vision) remains to be investigated.

**Broader Impact.** As a work centered around AIGC, it is probable that inappropriate contents may appear from the output. We should be aware of this negative societal impact.

