# OpenReview forum: "U-REPA: Aligning Diffusion U-Nets to ViTs"
_NeurIPS.cc/2025/Conference — NeurIPS 2025 poster_

### Official Review · Reviewer_U3kJ · 2025-06-29

**Clarity:** 3
**Significance:** 1
**Originality:** 2
**Rating:** 3
**Confidence:** 5

**Summary:**

This paper proposes U-REPA, which is an adaptation of Representation Alignment$^{[1]}$ for U-Net-based diffusion models. In this paper, the authors mainly focus on how to deal with the difference between DiT and U-Net, facing challenges when directly applying REPA. In my opinion, although this paper finds a better way to align representations on U-Net-based diffusion models, the novelty and contribution are still limited. So I lean to reject of this paper.

[1] Representation Alignment for Generation: Training Diffusion Transformers Is Easier Than You Think

**Questions:**

1. Make ablations on the loss term in Eqn.(5)

2. Provide the visualizations of the generated samples of REPA and U-REPA.

3. Can this method be applied on other U-Net-based diffusion models?

4. Compare U-REPA and REPA under the same experiment setting, especially the trained U-Net.

**Ethical Concerns:**

["NO or VERY MINOR ethics concerns only"]

**Final Justification:**

Although the author has addressed some doubts, given the novelty and contribution of this article, I will revise my score to Bordline Reject.

**Limitations:**

Although the authors propose U-REPA, an adaptation of REPA, which can boost the performance of REPA on U-Net-based diffusion models, but the novelty and contributions are limited. Others please see the weaknesses and questions.

**Paper Formatting Concerns:**

No paper fomatting concerns.

**Quality:**

3

**Strengths And Weaknesses:**

Strengths:

The paper explores a novel way to apply REPA on U-Net-based diffusion models, like SiTs and DiTs. When directly applying REPA on U-Net, the authors face several challenges and solve them properly:

1. Divergent block functionalities due to skip connections: aligning intermediate U-Net layers.

2. Spatial-dimension mismatches from U-Net downsampling: upscaling U-Net features and use post-MLP to solve spatial inconsistency.

3. Feature space gaps hindering tokenwise alignment: replacing cosine similarity loss with the manifold loss.


Weakness:

I hold the belief that the authors have properly solved the issues when they apply REPA on U-Net. My main concern is that contribution and novelty of this paper. Comparing with REPA, U-REPA seems to change the position of alignment, and change the alignment loss.

Besides, a interesting point I find is that the authors claim that directly apply REPA cannot work well, so they propose a new manifold loss. However, in Eqn.(5), I still find the whole loss consists of the original REPA loss. The authors do not make any ablation on it.

Besides, the authors do not provide any visualizations of the generated samples by using REPA and U-REPA.

The most important thing is that, the comparisons between REPA and U-REPA are unfair. In most of the tables, the U-Net used in U-REPA is better than the one used in REPA, this can be very strange. Why the authors do not put them under the same experiment settings?

---

> ### Author Rebuttal · Authors · 2025-07-31
>
> We sincerely thank reviewer U3KJ for their constructive comments. Our responses are as follows:
>
> ## Q1: Contribution & Novelty.
>
> The contribution of this paper goes beyond an engineering practice: it rethinks U-Net, which is a long-neglected architecture. Its most important contribution is that it demonstrates U-Nets perform no worse than the DiT-class.
>
> - To the best of our knowledge, we are the first to adapt REPA to U-Nets by making key modifications, including position, upsampling, and manifold loss.
> - There are some previous works that rethink U-Net (as cited in "Related Work"), but they fail to demonstrate U-Nets' best performance (in these work, usually >2 FID on ImageNet256). In contrast, we show U-Nets' **extreme best performance** is better than DiT at only half of the training cost. Our work reveals the great potential of U-Net, and hints a better architectural choice in diffusion.
>
> ## Q2: Ablation on the REPA loss.
>
> Thanks for your suggestions. To clarify, we don't reject the importance REPA loss (we will emphasize this point in the next revision); rather, we hold that REPA loss is reaching a bottleneck in the U-Net alignment setting and requires manifold guidance for assistance.
>
> In our setting, we follow the hyperparameter setting from REPA, which sets $\lambda=0.5$. We also enclose an ablation experiment table that reveals the importance of REPA loss while $\lambda$ is reduced to 0 and manifold loss is kept intact. Results indicate that REPA loss is important for generation quality:
>
> | $\lambda$        | 0.5  | 0.25 | 0 (No $\mathcal{L}_{REPA}$) |
> | ---------------- | ---- | ---- | --------------------------- |
> | FID $\downarrow$ | 5.72 | 6.42 | 10.91                       |
>
> ## Q3: Lack of Visualization Comparison.
>
> We have already put a visualization of our method in the supplementary materials due to space limits. Since we cannot attach a picture for comparison in the rebuttal, instead we conduct a human preference experiment that performs one-on-one comparison between REPA and U-REPA under the same sampling seed ($seed=0$) as follows.
>
> Our experiment consists of the first 100 images and no cherrypick is conducted. Due to content difference or similar quality, we find it hard to distinguish some image pairs. Results show the image quality of U-REPA is better than REPA, which coincides with the experiment results.
>
> | Which method is better? | REPA | U-REPA (Ours) | Can't Tell |
> | ----------------------- | ---- | ------------- | ---------- |
> | Counts                  | 17   | 38            | 45         |
>
>
> ## Q4: Compare U-REPA and REPA under the same experiment setting.
>
> To clarify, the whole process is started from scratch (according to the recipe of REPA);  we are not loading a trained U-Net for U-REPA (which might cause unfairness).
>
> While we have demonstrated that U-Net is more performant, this evaluation is only at initial phase (only @ 400K); we are not certain whether U-Net could achieve close SOTA performance. In our paper, we show that U-Net could also reach the comparable SOTA performance of DiT at a faster convergence speed under exactly the same, fair training recipe.

---

> ### Comment · Reviewer_U3kJ · 2025-08-07
> **Thanks for the rebuttal.**
>
> Thank you very much for the author's response to my issues. I noticed this paper a long time ago. But I will still insist that this paper falls between borderline reject and borderline accept.
>
> From my perspective, this paper mainly discusses how to apply REPA to the U-Nets based diffusion model. The method is not much different from REPA, but rather adapts to different network architectures, such as changing the position of alignment. In the final loss, REPA loss is not excluded.
>
> In addition, although the author has made some improvements in the U-Net based diffusion model compared to REPA, the mainstream framework now is the diffusion model of DiT architecture, so I have some doubts about the practicality of this paper.

---

> > ### Author Response · Authors · 2025-08-08
> > **Thanks for your response, and some clarifications**
> >
> > Dear Reviewer U3kJ,
> >
> > Thank you very much for your response! Here are our clarifications:
> >
> > ## About the practicality of REPA adapted to U-Net.
> >
> > We hold that effectively applying REPA to U-Net is important.
> >
> > Despite the current architectural trend of DiT, we are **debating U-Net is not unimportant** (as the original mainstream architecture) due to the following reasons:
> >
> > - The DiT paper claims "the U-Net inductive bias is not crucial to the performance of diffusion models", but **no supporting ablations** is presented in the DiT paper. The shift from U-Net to DiT takes place in academia not long ago; U-Nets are **neglected** in most recent research though they are not proved ineffective.
> >
> > - In U-REPA, we show that the U-Net architecture's extreme best performance is **better than DiT** with only half of the training iterations, giving a highly competitive architecture alternative.
> >
> > - As the previous mainstream choice, diffusion U-Net is still **widely used**. Before we conduct this work, we saw requests both on REPA's ICLR'25 review and in its GitHub issues enquiring whether REPA works on U-Net. Plus, in some downstream tasks like low-level diffusion task, U-Net **performs better** than DiT [1], hinting its useful applications.
> >
> > In a nutshell, while DiT has been the research focus of most recent papers, our U-REPA proposes (contrary to most beliefs in academia) that **U-Net is a diffusion architecture with lots of potentials**: when leveraged well, U-Nets could outperform DiTs in diffusion generation. While the diffusion community is converging to DiT as a lonely architectural choice, we think **alternative choices should be recalled for diversity** in the community.
> >
> > [1] Effective Diffusion Transformer Architecture for Image Super-Resolution. AAAI 2025.
> >
> >
> >
> >
> >
> > ## About the method difference with REPA.
> >
> > We hold that U-REPA and REPA are rather different: unlike DiT that has a homogeneous architecture (where changing the aligning position is merely changing the hyperparameter), U-Net has different stages with **different feature sizes**, thus requiring **tailored alignment designs**.
> >
> > We first show REPA at mid-layers is more optimal for models with skips, but since feature downsampling is involved at mid layers, **feature size difference** (between the diffusion model and the ViT encoder) has to be effectively addressed. Centering around this problem, we accordingly propose the way to **unify feature sizes** between U-Net and ViT-encoder, and introduce **manifold alignment loss** to fill space gaps of different-size features.
> >
> > Our paper also gives a bunch of **interesting discoveries** beyond the original REPA:
> >
> > - Via empirical analysis on the architecture, we hold that U-Net's **feature downsampling** is the key to U-Net's good performance (but this feature is also causing trouble to REPA, and we focus on addressing it in U-REPA).
> >
> > - Our U-Net analysis indicate that **semantic, outline-related block parts** are the key for conducting representation alignment. Unlike REPA where the early place for injection is merely a hyperparameter with very limited discussion, we hint that blocks with semantic functions require more visual encoder guidance.
> >
> > - Aligning high-freq, detail-rich ViT features with low-freq, detail-poor U-Net features results in a **similarity bottleneck** during training (they cannot be completely aligned by REPA's token-wise objective). Hence, besides REPA loss, we need other objectives for alignment.
> >
> > - U-Net achieves **even better performance than DiT** and **converges much faster** with the help of U-REPA. Though the current trend favors DiT, we hint that U-Net is possibly a better choice for diffusion generation.
> >
> > We hope that these interesting discoveries could be contributive to the community for future research.

---

> > > ### Comment · Reviewer_U3kJ · 2025-08-09
> > > **Thanks for the author`s response.**
> > >
> > > I understand that the contributions of this paper are the modifications of applying REPA to the U-Net based diffusion model, and I recognize it.
> > >
> > > However, I feel that the comparison method in the article lacks a certain degree of fairness. If the author wants to compare the performance of U-REPA and REPA, they should be compared in the same experimental setting, rather than using SiT for REPA and SiT $\downarrow$ for U-REPA. The performance of these two backbones is not consistent. How can you ensure that U-REPA is superior to REPA? Or will you also add tricks to train SiT$\downarrow$ and DiT$\downarrow$ as contributions for this paper?
> > >
> > > Prior to rebuttal, the authors did not demonstrate the performance differences between SiT and SiT$\downarrow$ in the text, nor did they conduct experiments on SiT$\downarrow$+REPA and SiT$\downarrow$+U-REPA. In the paper, Fig.3, Fig.4, as well as Tab.3, Tab.4, and Tab.5, all compare REPA and U-REPA under different conditions. Why did the author only provide such an important experiment during the rebuttal?
> > >
> > > Moreover, the paper lacks any visualization of generated images. How can the reviewers compare the differences between REPA and U-REPA generated samples? This kind of visualization is very common in the AIGC field. Why didn't the author add it?
> > >
> > > I found in the author's response Q1 to Reviewer # kqTd that the performance of SiT$\downarrow$ is much better than SiT, and the performance growth of SiT+REPA is much greater than that of SiT$\downarrow$+U-REPA, which has raised doubts about the experiments in this paper.
> > >
> > > If the author believes that they do need to conduct experiments on U-Net-based diffusion, but they cannot find the corresponding backbone and can only make modifications on SiT and DiT, why doesn't the author directly conduct experiments on DDPM[1], iDDPM[2], or even EDM[3], but choose to modify Transformer-based DiT and SiT?
> > >
> > > I still insist on my viewpoint that this paper alls between borderline reject and borderline accept.
> > >
> > > [1] Denoising Diffusion Probabilistic Models
> > >
> > > [2] Improved Denoising Diffusion Probabilistic Models
> > >
> > > [3] Elucidating the Design Space of Diffusion-Based Generative Models

---

> ### Author Response · Authors · 2025-08-09
> **Thanks for the reviewer's response and some further clarifications**
>
> Dear Reviewer U3kJ,
>
> We are sincerely grateful for your recognition of our contribution. After reading the response, we feel that there are quite a few misunderstandings. Here we hope to clarify them:
>
> ## C1,2: Fairly comparing between REPA and U-REPA.
>
> - We ensure U-REPA is superior to REPA by **comparing them on the same U-Net architecture** (which is what we have responded in "Kindly asking about further concerns"). This comparison is conducted in an entirely fair manner: all training, sampling, and architecture settings are exactly the same. The experiment data are quoted from the original U-REPA paper (we will clarify this issue in the third bullet point). Following previous practices, tricks are added to show the best potential of U-Net, which are not our contribution.
>
> - The reason why SiT results are omitted initially: In the original draft, we have **demonstrated the difference between DiT and DiT$\downarrow$ instead**. Since the SiT paper is an improvement on top of the DiT architecture, we thought **the trend of SiT / SiT$\downarrow$ would be very similar to DiT / DiT$\downarrow$,** which is verified by our experiments (shown in the following table). We hold that the advice of showing SiT performance makes sense and we will add SiT and SiT$\downarrow$ in the next paper revision.
>
> |     | DiT  | DiT$\downarrow$ | SiT  | SiT$\downarrow$ |
> | --- | ---- | --- | --- | --- |
> | FID$\downarrow$ | 19.5 | 11.0 | 17.2 | 9.2 |
>
> - **Actually we provided results of both SiT$\downarrow$+REPA and SiT$\downarrow$+U-REPA in the "Ablation" part of the paper; we only cited (not "provided") these data during the rebuttal.** SiT$\downarrow$+REPA (FID 9.35) is in line 2, Table 4 in the ablation study. We did not highlight this pair of comparison at first because we thought it was for sure that U-REPA performs much better than REPA on U-Net (otherwise this paper would be meaningless; the gap is indeed huge according to the results). But it turns out some confusion is taking place so we will highlight it in the next revision for better understanding.
> - We also hold that **comparing SiT+REPA and SiT$\downarrow$+U-REPA is meaningful** because we are **fairly conducting both experiments from scratch** and all settings are aligned. Via these experiments, we show beyond the effectiveness of U-REPA that **U-Net architectures have good potentials, both from performance and convergence.**
>
> ## C3: About Visualization
>
> We have provided a visualization **in the supplementary materials due to space limits**. We will move it back to the main paper in the next revision.
>
> **Why there is no difference comparison between REPA and U-REPA:** Actually, we are quite surprised by this request. From our knowledge, one-to-one comparison with other methods is **quite uncommon in most image generation methods** (e.g. DiT, SiT, REPA et cetera). But we are happy to provide one in the next revision. We have inspected image quality where U-REPA is clearly better. In other AIGC tasks where an image is inputted into the model, like Image-Editing or Super-Resolution, one-to-one visualization comparison is more common.
>
> ## C4: About Performance Growth
>
> We hold that **comparing SiT$\downarrow$ / SiT$\downarrow$ + U-REPA and SiT / SiT+REPA is unfair** due to the following reasons (we have also responded to BL2s in "Responses to 'Still missing comparison' ").
>
> - When generation performance is stronger, it is also harder to improve (especially for FID$\downarrow$).
> - Unlike SiT vs. ViT, SiT$\downarrow$ vs. ViT have key architectural differences, which is harder to align.
>
> Hence, we hold that this comparison is unfair: they cannot be simply measured by the amount of metric improvement. Comparing both REPA and U-REPA under SiT$\downarrow$ is fairer.
>
> ## C5: Why not Using Traditional U-Nets
>
> Our purposes of using DiT modification rather than [1,2,3] are as follows:
>
> - Firstly, through the modification of DiT, we want to segment U-Net components to evaluate their individual contribution and demonstrate the **importance of downsampling** in U-Net. This feature is important in our paper's effort in adapting REPA to U-Net: though downsampling is the key for U-Net's good performance, it is also the key barrier to the adaptation of REPA.
> - Secondly, these early U-Net architectures are designed for the **earlier, easier diffusion tasks and settings**. These tasks include smaller images and fewer class objects (e.g. ImageNet-64, CIFAR-10). [2] includes ImageNet-256, but its single-model performance only achieves FID 31.5 after 2400 epochs' training (>> 1400ep for DiT/SiT). **We doubt whether their capacity enables good generation performance.** In contrast, the setting of DiT is closer to current-day applications (it is more widely used and adapted, e.g. SiT and REPA). Therefore, we choose to use DiT as a popular choice for modification.
>
> Again, thank you for your advice and we will take it when revising our paper.
>
> Sincerely,
>
> Authors

---

### Official Review · Reviewer_kqTd · 2025-06-30

**Clarity:** 3
**Significance:** 4
**Originality:** 4
**Rating:** 5
**Confidence:** 4

**Summary:**

This paper introduces U-REPA, an adaptation of the REPA alignment methods specifically for U-Net style architectures. The main contributions include:

1. This paper suggests three key challenges in aligning features between diffusion U-Nets and ViT encoders: different block functionalities, spatial dimension inconsistencies, and larger feature space gaps.

2. Through experiments, the authors verify that the fast convergence advantage of U-Net primarily stems from multi-scale hierarchical modeling rather than skip connection. And a series of SiT$\downarrow$ models with modifications to U-Net style.

3. The proposed U-REPA, a new framework developed to address the challenges through intermediate layer alignment, feature size adjustment, and manifold loss.

4. Experiments demonstrate that U-REPA can achieve generation quality with FID < 1.5 on the ImageNet 256×256 dataset and outperforms REPA with only half the number of epochs under the same training settings.

**Questions:**

My questions are listed in the weakness part above. If the authors address my concerns after the rebuttal process, I would like to raise the final score.

**Ethical Concerns:**

["NO or VERY MINOR ethics concerns only"]

**Final Justification:**

I have read through all the reviews and author responses. I have discussed with the author during the phase and reached a final conclusion based on the rebuttal and discussion. The authors provide comprehensive explanations that have addressed all my main concerns. The new insight and supplement reinforce the quality and contribution of the paper. Thus, I raise the final rating from borderline reject to accept, which stands that the paper is ready for publication in the conference.

**Limitations:**

yes

**Paper Formatting Concerns:**

I have no formatting concerns about this paper.

**Quality:**

3

**Strengths And Weaknesses:**

**Strength**:

The paper proposes U-REPA, an effective representation alignment method for diffusion U-Nets that improves generation quality and training efficiency through intermediate layer alignment, feature upscaling, and manifold loss. The proposed architecture and alignment method can greatly improve the convergence and performance of image generation, supported by the extensive experiments.

**Weaknesses**:

1. Though the authors propose U-REPA to align features of the U-Net to ViT's, the alignment is conducted between a modified architecture SiT$\downarrow$ rather than the original convolutional U-Net. Experiments concerning the convergence and performance of the convolutional U-Net are necessary.

2. Despite the components of the macro components, skip connections and hierarchical design, the fast convergence of the original U-Net may come from the convolution modules. Discussion of convergence speed between convolutional nets and DiT should be included in this paper.

3. The writing of the paper should be checked and improved. In L208 of the main body, there exists a meaningless sentence ``The similarity gap between SiT and SiT$\downarrow$''. And the table, such as Table 1, is not cited in the main text. The symbol ${\textbf{h}^{\[n\]}_{t}}$ is not specified in the text.

4. It seems that the experiments are conducted only on global seed=0. Though this setting ensures the reproducibility, it still requires evaluating the quantitative experiment analyses and proposed methods across at least 3 random seeds, in order to avoid the fluctuation and cherry picking, and demonstrate robustness.

5. The Time-aware MLP may not be compatible with the main story of the paper, and the experiment shows that it has a minor contribution to the whole performance. Thus, in my opinion, it should be treated as a trick and should be removed from the main paper.

6. The proposed manifold loss is similar to the Marginal Distance Matrix loss in VA-VAE. It would be better to provide more discussion between the two losses in Section 3.3 and also refer to VA-VAE in that section, not just in the related work section.

---

> ### Author Rebuttal · Authors · 2025-07-31
>
> We sincerely thank reviewer kqTd for their constructive comments. Our responses are as follows:
>
> ## Q1&2: Regarding the convolutional U-Net.
>
> Unfortunately, we hold that the original conventional U-Net is rather limited in capacity and are not suitable for scaling up. We conducted experiments on SongUNet and figure out that it is not as performant as SiT$\downarrow$:
>
>
> |                        | 100K  | 200K  | 300K  | 400K  |
> | ---------------------- | ----- | ----- | ----- | ----- |
> | SiT+REPA\* (DiT architecture) | 19.7 | 11.9 | 9.4  | 9.0  |
> | SiT$\downarrow$+U-REPA | 12.8 |  6.6  |  6.2   | 5.4  |
> | SongUNet+U-REPA        | 29.0 | 18.8 | 14.9 | 12.9 |
>
> \* Replicated according to the official codebase and setting. In the official paper, SiT+REPA @ 400K is reported as 7.9 FID.
>
> We hold that the **capacity and property of architecture**, rather than **the use of convolution only**, is the key to fast convergence. SongUNet is a fused architecture of convolution and self-attention, but it fails to be converging faster than full transformer diffusion models.
>
> [1] Song, Yang, et al. Score-Based Generative Modeling through Stochastic Differential Equations. ICLR 2021.
>
>
>
>
> ## Q3: Writing should be improved.
>
> We apologize for the writing issues. The sentence in L208 should be "The similarity gap between SiT and SiT is obvious". Table 1 should be mentioned in the "Toy experiments on U-Net components" paragraph. $h_t^{[n]}$ refers to the n$^{th}$ token in an image. We will revise the paper in the next revision.
>
> ## Q4: Multiple seed for robustness.
>
> Thanks for your advice. To clarify, we didn't cherrypick seeds; we just used the identical training setting and codebase of REPA that adopted seed=0. We also add experiments when seed is 1 and 2 to verify the robustness of our method.
>
> We extend the training to 600K iterations (due to limited time and resources) and reach the results as follows:
>
> | SiT$\downarrow$-XL + U-REPA | seed=0 (Ours) | seed=1 | seed=2 | mean              |
> | --------------------------- | ------------- | ------ | ------ | ----------------- |
> | FID$\downarrow$             | 1.618         | 1.599  | 1.588  | 1.602 $\pm$ 0.012 |
>
> The experiments indicate that fluctuation is 0.01 at merely 600K iterations, which is tiny compared to the values.
>
>
>
> ## Q5: Time-aware MLP may not be compatible.
>
> Thanks for your suggestions. We agree that these part is a trick and minor. We will remove this part in the next revision. We are not using these tricks for the main experiments (as commented with the ablations).
>
> ## Q6: Discussion between VA-VAE loss and Manifold Loss
>
> Thanks for your suggestions. The key difference between the Marginal Distance Matrix loss in VA-VAE and the manifold loss are as follows:
>
> 1. Purpose Difference: in VA-VAE, loss is applied to VAE for better image compression; our manifold loss is applied to the model while the VAE is kept fixed based on the off-the-shelf sd-vae.
>
> 2. Form Difference: VA-VAE is a l1-norm-based loss while our loss is based on l2-norm. We also tried l1 as an alternative but found it harmful our setting, slightly degrading from FID 6.24 to 6.28.

---

> > ### Comment · Reviewer_kqTd · 2025-08-02
> >
> > Thank you for the detailed responses during the rebuttal phase, which have addressed my concerns. I have read through the author responses and the comments from other reviewers. Before my final rating decision, I still hold the same concern as Reviewer BL2s. From Table 8, we can see the effectiveness of U-REPA compared to the original REPA, as SiT↓-XL/2 + REPA is 6.25 FID and SiT↓-XL/2 + U-REPA is 5.72 FID. However, the raw performance of SiT↓-XL/2 is not demonstrated. For a comprehensive experiment and fair comparison, it should be included in this paper.

---

> > > ### Author Response · Authors · 2025-08-03
> > > **Responses to "Official Comment by Reviewer kqTd"**
> > >
> > > We sincerely thank reviewer kqTd for their constructive responses and suggestions in the discussion phase.
> > >
> > > ## Q1: The performance of SiT$\downarrow$-XL/2.
> > >
> > > We agree that the SiT$\downarrow$ performance is important. We add the raw performance of SiT$\downarrow$-XL/2 in the following table, and we will include it in the next revision:
> > >
> > > | Method                           | FID$\downarrow$ |
> > > | -------------------------------- | --------------- |
> > > | SiT-XL                           | 17.2            |
> > > | SiT-XL+REPA                      | 7.9             |
> > > | SiT$\downarrow$-XL               | 9.2             |
> > > | SiT$\downarrow$-XL+U-REPA (Ours) | 5.4             |
> > >
> > > ## Q2: The advantage of SiT$\downarrow$+U-REPA over SiT$\downarrow$+REPA.
> > >
> > > We hope to clarify that the FID of SiT$\downarrow$+REPA is 9.35 in the ablation setting (Table 4, because the original REPA performs matching of features of the same size at early layer). The cfg comparison between SiT$\downarrow$+U-REPA and SiT$\downarrow$+REPA is thus as follows:
> > >
> > > | Method                           | FID$\downarrow$ |
> > > | -------------------------------- | --------------- |
> > > | SiT$\downarrow$-XL+REPA          | 9.35            |
> > > | SiT$\downarrow$-XL+U-REPA (Ours) | 5.72            |
> > >
> > > where the proposed U-REPA is far more effective for SiT$\downarrow$.

---

> > > > ### Comment · Reviewer_kqTd · 2025-08-03
> > > > **Rebuttal Followup**
> > > >
> > > > Thank you for the subsequent ablation result and the clarification. I miss the point that the original REPA performs matching at an early layer. It would be better to clarify that Table 4 experiments with REPA to choose the optimal alignment layer. And Table 8 improves the alignment with U-REPA at the optimal layer.
> > > >
> > > > I would like to raise my final rating toward acceptance. Please ensure that the revised version includes rebuttal details and clarifications. Good luck!

---

> > > > > ### Author Response · Authors · 2025-08-03
> > > > > **Thank You**
> > > > >
> > > > > We will include them in the next revision. Thank you so much!

---

### Official Review · Reviewer_i6qu · 2025-07-01

**Clarity:** 3
**Significance:** 3
**Originality:** 3
**Rating:** 4
**Confidence:** 3

**Summary:**

This manuscript aims to develop representation alignment method for UNet-based diffusion models. The authors perform experiments and point out that the challenges of this lies in the layer selection, dimension difference,  and feature compatibility. To this end, the authors take corresponding designs and propose to apply manifold loss for better alignment effect. Results show that the proposed U-REPA achieves better performance than vanilla REPA for UNet-based models.

**Questions:**

The author is recommended to optimize the layout to make it better.

**Ethical Concerns:**

["NO or VERY MINOR ethics concerns only"]

**Final Justification:**

The rebuttal has addressed part of my concerns. I am inclined to accept this paper.

**Limitations:**

yes

**Quality:**

3

**Strengths And Weaknesses:**

- Strengths:
1. The proposed method is well-motivated and the analysis is sufficient.
2. Extensive experiments demonstrate that the proposed U-REPA achieves great performance.
3. The analysis and ablation experiments are sufficient and reasonable.

- Weaknesses:
1. This work mainly involves some engineering analysis and engineering practice. The methods used, such as manifold loss, have also been proposed.
2. This work is just a extension of REPA for UNet-based framework with some  engineering conclusions and do not give new findings for promoting AIGC fields.

---

> ### Author Rebuttal · Authors · 2025-07-31
>
> We sincerely thank reviewer i6qu for their constructive comments. Our responses are as follows:
>
> ## Q1&2: Clarifying our core contributions.
>
> Our core contributions are as follows:
>
> 1. While DiT is becoming the mainstream architecture and U-Net is gradually being phased out, prior work has shown that U-Net converges quickly, which is an exploitable advantage. However, its upper bound in terms of generation quality and convergence speed remains unexplored (previous U-Nets report FID>2, which is still a large gap from SOTA). We aim to investigate the full potential of U-Net in these aspects (Finally achieving 1.41 FID at merely 1M iterations).
>
> 2. A plain use of REPA does not work well on U-Net due to architecture mismatch. We adapt REPA to U-Nets by making key modifications, including position, upsampling, and manifold loss.
>
> In a nutshell, beyond proposing a method, this paper shows that U-Net is perhaps **a better choice of diffusion architecture** both in terms of convergence and generation quality, paving the way for future research.
>
>
>
> ## Q3: Optimizing the layout is recommended.
>
> We are sorry that our layout is too cramped due to space limits. We will move some less important part to the appendix part for better clarity and readability.

---

> ### Author Response · Authors · 2025-08-09
> **Further Emphasizing the contribution of our paper to the AIGC community**
>
> Beyond the original REPA, our paper also gives a bunch of **interesting discoveries** as follows:
>
> - Via empirical analysis on the architecture, we hold that U-Net's **feature downsampling** is the key to U-Net's good performance (but this feature is also causing trouble to REPA, and we focus on addressing it in U-REPA).
>
> - Our U-Net analysis indicate that **semantic, outline-related block parts** are the key for conducting representation alignment. Unlike REPA where the early place for injection is merely a hyperparameter with very limited discussion, we hint that blocks with semantic functions require more visual encoder guidance.
>
> - Aligning high-freq, detail-rich ViT features with low-freq, detail-poor U-Net features results in a **similarity bottleneck** during training (they cannot be completely aligned by REPA's token-wise objective). Hence, besides REPA loss, we need other objectives for alignment.
>
> - As is stressed, U-Net achieves **even better performance than DiT** and **converges much faster** with the help of U-REPA. Though the current trend favors DiT, we hint that U-Net is possibly a better choice for diffusion generation.
>
> We also hope that these interesting discoveries could be contributive to the AIGC community for future research.

---

### Official Review · Reviewer_BL2s · 2025-07-01

**Clarity:** 2
**Significance:** 3
**Originality:** 3
**Rating:** 5
**Confidence:** 4

**Summary:**

This paper addresses the problem of adopting representation alignment (REPA) from the DiT to the diffusion U-Net. Despite U-Net-based diffusion models being known to converge faster than DiT, the authors found that adopting REPA to the U-Net architecture is not trivial due to (i) different functionalities of U-Net blocks, (ii) spatial-dimension inconsistencies from U-Net, and (iii) space gaps between U-Net and ViT. To address this issue, the authors propose U-REPA that aligns ViT features to the middle stage of U-Net with an upsampling strategy and incorporates a manifold loss that regularizes the relative similarity between samples. Experimental results show that U-REPA consistently and significantly outperforms REPA.

**Questions:**

1. Please answer the weaknesses.
2. 1. Could the authors provide the SiT$\downarrow$-XL/2 results in Table 4?

**Ethical Concerns:**

["NO or VERY MINOR ethics concerns only"]

**Final Justification:**

After reading the authors' rebuttal, my concerns are well addressed. Therefore, I would like to raise my final rating to "Accept".

**Limitations:**

The authors have not included the limitations and potential negative societal impact of the work, e.g., the societal impact of image generation models (deepfake, image copyright).

**Quality:**

2

**Strengths And Weaknesses:**

**Strengths**
1. The paper is overall easy to follow.
2. Toy experiments demonstrate the potential of U-Net-based diffusion models, further motivating the necessity of applying REPA for diffusion U-Net.
3. Comprehensive analysis of adopting REPA on diffusion U-Net.

**Weaknesses**
1. Important but missing comparison: In this paper, there is no comparison between SiT$\downarrow$ and SiT$\downarrow$ + U-REPA. While the proposed alignment scheme for the diffusion U-Net is intriguing, I am concerned whether the alignment loss is indeed beneficial for the diffusion U-Net architecture due to the lack of a comparison between SiT$\downarrow$ and SiT$\downarrow$ + U-REPA. For instance, DiT-XL/2 already outperforms DiT-XL/2+REPA at 400K iterations (DiT-XL/2 gets 11.02 FID scores vs DiT-XL/2+REPA gets 12.3 FID score.).
2. Could the authors provide more analysis of incorporating the manifold alignment loss? I wonder whether the manifold alignment loss affects token-wise similarities. While SiT achieves larger token-wise similarity than SiT$\downarrow$, I believe SiT$\downarrow$ gets better generation quality than SiT. Based solely on these results, I think the direct alignment using L2 loss between the feature vectors is not beneficial for the U-Net.
3. Because the proposed method includes an upsampling layer for alignment, I think a higher resolution experiment (e.g., ImageNet 512x512) is necessary.
4. The font size in Figure 2 is too small to understand.

---

> ### Author Rebuttal · Authors · 2025-07-31
>
> We sincerely thank reviewer BL2s for their constructive comments. Our responses are as follows:
>
> ## Q1: Important but Missing Comparison.
> We hope to clarify that the value 11.02 FID is in fact for DiT$\downarrow$-XL, the performance **our proposed U-Net architecture (instead of SiT-XL that is based on a conventional DiT architecture)**. Below we provide a table illustrating the comparison.
>
> | Model | FID @ 400K |
> | ------------------ | ---- |
> | DiT-XL             | 19.47 |
> | SiT-XL+REPA        | 7.9 |
> | DiT$\downarrow$-XL | 11.02 |
> | SiT$\downarrow$-XL+U-REPA (Ours) | 5.4 |
>
> From the table, it is obvious that both REPA and U-REPA could boost the generation performance.
>
> ## Q2: More analysis of incorporating the manifold alignment loss.
>
> Yes definitely. The manifold alignment loss is imposing very small amount of change on tokenwise cosine similarities (unfortunately, we cannot put a plot here to illustrate). Whether adding manifold loss or not, the tokenwise cosine similarity will always fluctuate around 0.6 (which we believe is the upper bottleneck for the alignment between U-Net and ViT encoders). In spite of this, manifold loss is adding regularization on the manifold aspect and thus making U-Net more performant.
>
> ## Q3: Higher-Resolution (ImageNet 512x512) Experiments.
>
> Thanks for your suggestion! We are happy to provide the performance of U-REPA on ImageNet 512x512 in the table below. U-REPA outperforms REPA by considerable margins, showing that the proposed U-REPA could also be generalized to larger images. Due to time and resource constraints (running ImageNet-512 is very resource intensive), we could reach 400K iterations at this moment.
>
>
> | FID$\downarrow$  / IS $\uparrow$ | SiT$\downarrow$-XL/2+REPA | SiT$\downarrow$-XL/2+U-REPA |
> | -------------------------------- | ------------------------- | --------------------------- |
> | 400K                             | 2.44 / 247.3              | 2.21 / 274.7                |
>
>
> ## Q4: Font size is too small.
>
> We are terribly sorry for the overly-small fontsize in the paper. We will enlarge it in the next revision for better readability.

---

> > ### Comment · Reviewer_BL2s · 2025-08-02
> > **Still missing comparison (Q1)**
> >
> > **[Q1]** I think it is not indeed obvious to say U-REPA boosts or improves the generation quality: What is the performance of SiT$\downarrow$-XL? Is it worse than SiT$\downarrow$-XL + U-REPA? DiT vs SiT is unfair comparison. Because DiT$\downarrow$-XL already achieves better performance than DiT-XL + REPA, I am still concerned whether U-REPA is indeed good alignment or not from comparison DiT / SiT+REPA vs DiT$\downarrow$ / SiT$\downarrow$ + U-REPA.

---

> > > ### Author Response · Authors · 2025-08-03
> > > **Responses to "Still missing comparison"**
> > >
> > > Again, we sincerely thank reviewer BL2s for their constructive responses and suggestions in the discussion phase.
> > >
> > > ## Q1[A]: The performance of SiT$\downarrow$-XL.
> > >
> > > Thanks for raising this issue and we are very sorry for not providing SiT$\downarrow$-XL results at the first rebuttal phase (we had very limited resource at that time). From the table below, we can see that U-REPA has a clear improvement on SiT$\downarrow$.
> > >
> > > | Method                    | FID$\downarrow$ |
> > > | ------------------------- | --------------- |
> > > | SiT-XL                    | 17.2            |
> > > | SiT-XL+REPA               | 7.9             |
> > > | SiT$\downarrow$-XL        | 9.2             |
> > > | SiT$\downarrow$-XL+U-REPA | 5.4             |
> > >
> > > Based on this table, we also hope to clarify that both DiT$\downarrow$-XL and SiT$\downarrow$-XL perform **worse** than SiT-XL+REPA (REPA is based on SiT so there is no DiT+REPA).
> > >
> > > ## Q1[B]: Whether U-REPA is indeed good alignment.
> > >
> > > We think it is quite **hard** to fairly compare between DiT / SiT+REPA vs. DiT$\downarrow$ / SiT$\downarrow$ + U-REPA for better alignment the following reasons:
> > >
> > > 1. As generation performance gets stronger, it is also becoming **much harder** to improve (especially for FID when it gets lower).
> > >
> > > 2. Aligning SiT$\downarrow$ and ViT is **much harder** than aligning SiT and ViT, because the backbone of SiT and ViT encoders are very similar. Aligning SiT$\downarrow$ to ViT encoder is a special case due to great architecture difference.
> > >
> > > As an alternative, we hold that SiT$\downarrow$ + U-REPA could be compared with SiT$\downarrow$ + REPA because they are performed on the same pair of models. We demonstrate the comparison in the table as follows (under the ablation setting):
> > >
> > > | Method                           | FID$\downarrow$ |
> > > | -------------------------------- | --------------- |
> > > | SiT$\downarrow$-XL+REPA          | 9.35            |
> > > | SiT$\downarrow$-XL+U-REPA (Ours) | 5.72            |

---

> > > > ### Comment · Reviewer_BL2s · 2025-08-03
> > > > **Official comment**
> > > >
> > > > Thank you for the responses with additional experiments. My concerns are well-addressed, and thus, I would like to raise my scores to "Accept".

---

> > > > > ### Author Response · Authors · 2025-08-03
> > > > > **Thank You**
> > > > >
> > > > > Thank you so much! We will include the experiment results in the next revision.

---

### Author Response · Authors · 2025-08-09
**Summary of the rebuttal and discussion phase**

We thank all reviewers for their constructive comments. Here we summarize the rebuttal and discussion with reviewers about key concerns as follows:

## What is the SiT$\downarrow$ performance?


We conduct experiments on SiT$\downarrow$ as follows:

|                 | DiT  | DiT$\downarrow$ | SiT  | SiT$\downarrow$ |
| --------------- | ---- | --------------- | ---- | --------------- |
| FID$\downarrow$ | 19.5 | 11.0            | 17.2 | 9.2             |

we did not include this result in the paper draft not because we want to conceal the results; it's because have included DiT/DiT$\downarrow$ rather than SiT/SiT$\downarrow$. Following reviewers' advice, we hold that SiT$\downarrow$ is important and will include it in the next revision.


## How to ensure fair comparison?

We hold that comparing SiT$\downarrow$+REPA and SiT$\downarrow$+U-REPA is fair under exactly the same architecture and settings. The results are as follows: (they are all cited from the "Ablation Study" in the paper; we did not provide these data in the rebuttal)

| Method                           | FID$\downarrow$ |
| -------------------------------- | --------------- |
| SiT$\downarrow$-XL+REPA          | 9.35            |
| SiT$\downarrow$-XL+U-REPA (Ours) | 5.72            |

We hold that it is unfair to compare the improvement SiT$\downarrow$ / SiT$\downarrow$ + U-REPA and SiT / SiT+REPA because:

- When generation performance is stronger, it is also harder to improve (especially for FID$\downarrow$).

- As U-Net is significantly different from ViT, the difficulty of aligning U-Net to ViT encoders and DiT to ViT encoders is different.

## Why not using traditional U-Nets?

Because traditional U-Net is limited in capacity; they are mainly for simpler tasks (typically smaller image, fewer classes).

The setting of DiT is related to more current-day applications and it is also inherited by REPA. Thus we mainly adopt their setting rather than the traditional U-Net.

## Our method contribution.

Our method targets at REPA's adaptation to U-Net. In contrast to DiTs, U-Nets have the unique design of down-sampled stage in the middle of the model; but middle of the model is also the best place for representation alignment in models with skips. Hence, tailored feature alignment between different sizes is required. We introduce size alignment and manifold space alignment loss to align U-Nets with ViT encoders.

Our method also gives several interesting discoveries which we hope could inspire future discoveries [(post)](https://openreview.net/forum?id=im3FJ6quii&noteId=y3yoJE6nlU), which we hope could contribute to the community for future research. In a nutshell, our paper shows that U-Net has great potentials in terms of both performance and convergence.



We will carefully consider all advice from reviewers and add clarifications and experiments in the next revision.

---

### Note · Authors · 2025-08-13

Dear Reviewers and AC,

Again, we sincerely thank the reviewers for their suggestions, helping us improve this paper. Here we provide final remarks of the rebuttal and discussion phase.

We have tried our best to answer questions from reviewers and resolve their concerns. We supplemented experiments, highlighted certain experiments in the paper, and provided clarifications. After the discussion, **Reviewer BL2s, i6qu, and kqTd have been convinced and incline towards acceptance**.

**Reviewer U3kJ initially hold that the paper lies between score "borderline accept" and "borderline reject"** after the rebuttal. In the first round of discussion, we resolved their concerns about **the contribution and practicality of this paper,  receiving their recognition**. In the second round of discussion, some more concerns from our discussion phase with other reviewers were raised. We found there were **quite a few misunderstandings** and we managed to clarify them in the last reply. Unfortunately, we are unable to receive further feedback due to the deadline.

For major concerns, we have presented a detailed summary of our responses in the "**Summary of the rebuttal and discussion phase**" comment.

At the very end of the discussion, we want to express our **sincere gratefulness** to the reviewers. They actively participate in the rebuttal discussions and help us improve this paper. **We will take their advice in the next revision**, including adding all rebuttal experiments, highlighting certain experiments in the paper, and including all clarifications. We hope this paper would **contribute to the AIGC community**, not only about the method, but also hinting great potential of U-Nets and offering architectural alternatives other than the converged community design of DiT.

Sincerely,

Authors

---

### Decision · Program_Chairs · 2025-09-17

**Decision:**

Accept (poster)

**Comment:**

Final rating: 5:Accept/ 4: Borderline Accept/ 5:Accept /3:Borderline Reject. This paper introduce U-REPA, extending REPA to diffusion U-Nets by addressing block heterogeneity, downsampling mismatches, and ViT–U-Net token gaps. It aligns at the U-Net mid-stage, upsamples MLP-projected features, and replaces token-wise similarity with a manifold loss. Experiments show faster convergence and FID < 1.5 on ImageNet 256×256 in 200 epochs (1M iters), outperforming REPA with ~½ the training under sd-vae-ft-ema.

Reviewers acknowledge the strong experiments and ablations but raise concerns about the missing comparison between SiT↓ and SiT↓+U-REPA, absent results with a convolutional U-Net, and other missing details and discussion. After the rebuttal, the concerns of reviewers BL2s, i6qu, and kqTd were largely addressed and they now lean toward acceptance; however, reviewer U3kJ still questions the fairness of key experiments demonstrating U-REPA’s contribution, notes missing experiments on SiT↓+REPA and SiT↓+U-REPA, points out the lack of direct visual comparisons between REPA and U-REPA, and requests additional runs on DDPM, iDDPM, or EDM.

The ACs conclude that fairness is ensured by comparing methods on the same U-Net architecture; they also note that the paper already contains related experiments for SiT↓+REPA and SiT↓+U-REPA, judge the absence of visual side-by-sides as non-critical given strong quantitative results (though encouraged for completeness), and consider that the DDPM/iDDPM/EDM request misaligned and  does not make sense given that those works are actually about diffusion sampling operation but reviewer U3kJ's concern is about vision backbone architectures.

Overall, the ACs find the rebuttal satisfactory, view the paper as a strong contribution to feature alignment in U-Nets—highlighting their potential and offering an alternative to the prevailing DiT paradigm—and recommend acceptance.